# Factorial structure of the patient health questionnaire-9, generalized anxiety disorder-7 and berger HIV stigma scale-short form among adolescents living with HIV in Ghana

**Samuel Adjorlolo**[1,2]*, **Dorothy Serwaa Boakye**[3]

**1** Department of Mental Health Nursing, School of Nursing and Midwifery, University of Ghana, Legon, Ghana, **2** Research and Grant Institute of Ghana, Legon, Ghana, **3** Department of Health Administration and Education, University of Education, Winneba, Ghana

* sadjorlolo@ug.edu.gh

## Abstract

### Background

Mental health issues and stigma experiences significantly affect adolescents living with HIV (ALHIV). While the Patient Health Questionnaire-9 (PHQ-9), Generalized Anxiety Disorder-7 (GAD-7), and Berger HIV Stigma Scale-Short Form (HSS-SF) are widely used screening tools for mental health and stigma, their factorial validity among ALHIV in resource-limited settings such as Ghana remains understudied. This study investigated the factor structure, internal consistency, and correlations of the PHQ-9, GAD-7, and HSS-SF among ALHIV in Ghana.

### Methods

A cross-sectional study was conducted among 105 ALHIV (aged 10−24 years) receiving care at three government hospitals in Eastern Ghana. Confirmatory factor analyses were conducted to examine the factor structure of the PHQ-9 and GAD-7 and HSS-SF.

### Results

A two-factor structure showed the best fit for PHQ-9, with satisfactory internal consistency for cognitive/affective (α = 0.79) and somatic factors (α = 0.70). The GAD-7 demonstrated best fit as a unidimensional factor structure, with acceptable internal consistency (α = 0.88). The HSS-SF was found to be best represented by a three-factor structure, with internal consistency ranging from 0.76 to 0.95 across subscales.

### Conclusions

The PHQ-9, GAD-7, and HSS-SF demonstrated acceptable factorial structure and internal consistency among Ghanaian ALHIV, supporting their utility as screening tools in this population.

**Data availability statement:** Data is available as "Mental health burden in adolescents living with HIV", Mendeley Data, V2, doi: 10.17632/g2n474d3pt.2

**Funding:** The study was made possible (in-part) by the Preparing Outstanding Social Science Investigators to Benefit Lives and Environments in Africa Initiative (POSSIBLE-Africa) (Grant NO# POS-24-07), an initiative of the Science for Africa Foundation (SFA Foundation) enabled by the support of Carnegie Corporation of New York. The statements made and the views expressed are solely the responsibility of the authors and necessarily those of the SFA Foundation and partners. The funders had no role in study design, data collection and analysis, decision to publish, or preparation of the manuscript.

**Competing interests:** The authors have declared that no competing interests exist.

## Introduction

Mental health challenges and experiences of stigma are significant concerns among adolescents living with HIV (ALHIV), globally [1]. These issues are particularly acute in resource-limited settings of sub-Saharan Africa (SSA), where underdeveloped mental health systems, pervasive poverty and limited access to mental health services create substantial barriers to interventions to promote psychological well-being [2–4]. Early identification of mental health problems and stigma experiences among ALHIV is crucial for implementing timely interventions and improving psychosocial outcomes in this vulnerable population [5]. Historically, structured or semi-structured clinician-administered interviews have served as the gold standard for diagnoses of mental health disorders [6]. However, their implementation in resource-limited settings faces significant challenges. For example, these assessment methods require substantial time, expertise, and resources, making them impractical for routine screening in SSA context [7,8].

To obviate these limitations, several self-report measures have been developed for use among different populations, including adolescent population. Notable among them are the Patient Health Questionnaire-9 (PHQ-9), Generalized Anxiety Disorder-7 (GAD-7) and Berger HIV Stigma Scale (HSS). These measures have the potential to identify at-risk individuals for a more detailed clinical assessment. This will not only save cost, but more crucially, their use will facilitate the screening of ALHIV within the shortest possible time, and with limited resources. In view of this, the PHQ-9, GAD-7 and Berger's HSS are expected to demonstrate sound psychometric properties not only in settings where they have been developed, but more so in settings where they are being administered. However, as discussed below, there are concerns about the factor structure and internal consistency of the measures, hence the focus of the study.

The PHQ-9 is a 9-item tool commonly used to assess the presence and severity of depressive symptoms. Although the PHQ-9 was developed based on the Diagnostic and Statistical Manual of Mental Disorders (DSM) 4th edition (DSM-IV) [9], it is still theoretically consistent with the DSM V [10]. Developed on primary care sample, the PHQ-9 has been applied among the general population, including adolescents in school [8]. The PHQ-9 has demonstrated high internal consistency in multiple studies across different countries and populations [6,11–13]. However, there are conflicting findings regarding its factor structure, with evidence for both one-factor structure [8,14] and two-factor structure [6,15]. Among the HIV population, the PHQ-9 was found to demonstrate a one-factor [16].

The GAD-7 was developed as a brief, seven-item questionnaire that reflects the symptom descriptions of generalized anxiety disorder based on the DSM-IV [17]. However, it has been administered to screen for several anxiety disorders such as panic disorder and social anxiety [17,18], with good internal consistency[6, 7]. Like the PHQ-9, the findings relating to the factor structure of the GAD-7 are mixed, with some studies reporting one- factor [7,19–21] while others found evidence for a two-factor structure [6,22]. Among HIV population, the GAD-7 demonstrated one-factor [23,24].

The Berger HIV Stigma Scale-short form (HSS-SF) was developed from the Berger HIV Stigma Scale (HSS) [25] by Wright and colleagues as a brief tool to screen for stigma experiences among young people living with HIV [26]. The Berger HSS is one of the most widely used stigma scale across countries, with good psychometric properties [27]. Using exploratory factor analysis, Wright and colleagues reported a four-factor structure of the Berger HSS-SF as follows: Personalized Stigma, Disclosure Concerns, Negative Self Image, and Public Attitudes [26]. The HSS-SF demonstrated acceptable internal consistency, with reliability coefficients ranging from 0.72 to 0.84 [26]. However, unlike the PHQ-9 and GAD-7 whose psychometric properties have been extensively studied, the HSS-SF has not undergone confirmatory factor analysis to validate its factor structure. This gap in validation limits its broader applicability in different cultural and demographic contexts, particularly among ALHIV in resource-limited settings.

### The present study

The burden of mental health and stigma among ALHIV in SSA provides enough impetus for mental health services to be delivered to this population. This includes screening to detect cases for detailed mental health assessment and to inform intervention programming. In high-resource settings, routine screening for depressive symptoms has been integrated into HIV management protocols to improve mental health outcomes among people living with HIV/AIDS [28]. Similar initiatives are emerging in Ghana, where the government and international partners are working to integrate mental health services into primary care for ALHIV and other patient population.

The implementation of standardized screening tools like the PHQ-9, GAD-7, and Berger HSS-SF could significantly enhance this integration effort by providing systematic assessment of mental health challenges and stigma experiences. However, the effective implementation of these screening tools requires careful consideration of their psychometric properties, particularly in new cultural contexts. A significant concern relates to the inconsistent findings regarding the underlying factor structure. These measures are either unidimensional and/or multidimensional, suggesting a somewhat different conceptualization by previous studies. While a multidimensional factor structure connotes the existence of independent but inter-correlated groups of symptoms, unidimensional factor structure reflects homogeneity of symptoms [6]. The lack of clarity on the factor structure of the above measures is worrying, particularly when they are administered in settings other than where they have been validated such as Ghana. These structural inconsistencies raise important questions about the cross-cultural validity of these measures. The manifestation and assessment of mental health symptoms and stigma experiences are significantly influenced by sociocultural and geopolitical factors [29,30]. This influence is particularly relevant in Ghana, where cultural perspectives on mental health and stigma appear to differ from the contexts in which these measures were originally developed and validated. However, only a few studies have investigated the factor structure of PHQ-9, GAD-7 and Berger HSS-SF among people living with HIV in SSA [16,23], with none conducted among ALHIV in Ghana.

To address these gaps, this study aimed to investigate the factor structure of the PHQ-9, GAD-7 and Berger HSS-SF, and determine their internal consistencies and correlations among ALHIV in Ghana. By doing so, the study seeks to enhance the accuracy and efficiency of mental health and stigma screening in contexts where comprehensive clinical assessments may not be feasible. The findings will contribute to multiple areas including: (1) expanding the available screening resources for mental health and stigma assessment in Ghana; (2) deepening the theoretical understanding of how these constructs manifest in this specific population; (3) informing the practical implementation of screening programs in similar resource-limited settings; and (4) advancing scientific discourse on the cross-cultural assessment of behavioral health challenges.

## Methods and materials

### Study design and setting

The study employed a cross-sectional research design and was conducted at the Atua Government Hospital, the Akuse Government Hospital and the Asesewa Government Hospital. The selection of these facilities was strategic given their

age-long role in providing healthcare services to a significant proportion of people living with HIV/AIDS in two of Ghana's most affected districts: Lower and Upper Manya Krobo Districts. The Districts have a high HIV prevalence rate of 5.6%, which significantly exceeds both the regional average of 2.8% and the national average of 1.6% [31,32]. About 60% of the population in the Districts is under the age of 25 years [33]. Each facility maintains dedicated HIV clinics with specialized staff and serves between 200–500 HIV patients monthly, of whom approximately 15–20% are adolescents.

## Study population and sample size

The study population included adolescents (aged 10–24 years) living with HIV/AIDS who were receiving care at the healthcare facilities mentioned above. Healthcare providers at each facility assisted in identifying potential participants who met the inclusion criteria, and the research team approached those who expressed interest in participating in the study. The eligibility criteria were: Age 10–24 years; clinical diagnosis of HIV; enrolled in HIV care at any of the selected healthcare institutions; awareness of HIV status for at least six months; and able to provide informed consent and assent (and parental consent where applicable). Adolescents were excluded from participation if they were critically ill, have severe cognitive impairment that would affect their ability to participate in the study, are living in institutional care rather than with family, or are unable to communicate in either English or the local language (Dangbe). They were recruited using a convenient sampling approach. The distribution of participants across the facilities is as follows: Atua Government Hospital = 63, Akuse Government Hospital = 16, and Asesewa Government Hospital = 26 with an overall sample size of 105.

Of the total of 120 adolescents living with HIV in Atua, only 75 within the required age range fully satisfied the inclusion criteria (i.e., awareness of HIV status for at least 6 months). Out of this, only 63 provided informed consent and assent and were available for data collection, representing 60% of the final sample. In Akuse, a total of 55 ALHIV were eligible; however, only 20 had an awareness of their HIV status for at least 6 months. Out of these,16 consented and participated in the data collection, representing 15% of the final sample. In Asesewa, a total of 72 ALHIV were eligible with only 30 satisfying the criteria of awareness of HIV status for at least 6 months. Out of these, 26 provided informed consent and assent and participated in the study, representing 25% of the final sample. The final sample size of 105 participants (63 + 16 + 26) was therefore determined by the number of eligible adolescents who both met the inclusion criteria and provided informed consent and/or assent across all three study locations.

## Data collection measures

The main language spoken in the study site was Dangbe. To facilitate the data collection process, all the instruments underwent forward and backward translation from English to Dangbe following the WHO's translation guidelines to ensure cultural and linguistic equivalence [34]. The translation procedure followed a rigorous protocol as recommended by WHO. Two independent bilingual translators, both native Dangbe speakers with proficiency in English, completed the forward translation of the original English questionnaires. Both translators were familiar with healthcare terminology and the cultural context of ALHIV in Ghana. The research team and the translators then met to compare the two forward translations, resolved any discrepancies, and created a reconciled version that best captured the intended meaning of each item while maintaining cultural appropriateness. This reconciled version was back-translated into English by two different bilingual translators who had no prior knowledge of the original questionnaires. These translators were fluent in English and Dangbe. An expert committee comprising two healthcare professionals specializing in adolescent HIV care, psychologists, the first author, and the translators reviewed all translations, assessing semantic, idiomatic, experiential, and conceptual equivalence between the source and target versions and resolved any discrepancies. The translated questionnaires were pre-tested with a sample of 9 adolescents living with HIV (not included in the main study) to identify any items that were difficult to understand or culturally inappropriate. Participants were interviewed about their understanding of each question to ensure clarity and relevance. Based on the feedback from this cognitive debriefing, final adjustments were made to the questionnaires, and the expert committee approved the final versions for use in the study. This comprehensive translation

procedure ensured that the questionnaires maintained their psychometric properties while being culturally and linguistically appropriate for the study population in the Dangbe-speaking communities.

*Patient Health Questionnaire-9 (PHQ-9)* is a widely validated screening tool for depressive symptoms [35]. The PHQ-9 consists of nine items rated on a 4-point scale (0 = not at all to 3 = nearly every day), with total scores ranging from 0 to 27. The PHQ-9 scores were computed by totaling the responses from all nine items.

*Generalized Anxiety Disorder-7 (GAD-7)* was used to assess anxiety symptoms [17]. This 7-item instrument uses a 4-point rating scale (0 = not at all to 3 = nearly every day), with total scores ranging from 0 to 21. The total GAD-7 score was calculated by summing the scores for each of the 7 items.

*Berger HIV Stigma Scale-Short Form (HSS-SF)* was administered to screen for stigma experiences. The HSS-SF was adapted from the Berger HSS. It consists of 10-items rated using a 5-point Likert response format. Total scores are obtained by adding the items, with higher score reflecting more stigma experiences. The data collection measures are attached as supplementary materials.

## Data collection procedure

Participant recruitment and data collection commenced on 31st October 2024 and ended on 18th November 2024. The recruitment process began with healthcare providers inviting individual potential participants to join the study. Interested individuals were scheduled for initial meetings at their respective healthcare facilities where they were screened for eligibility. Additional participants were recruited during routine healthcare visits if they expressed interest in participating in the study. The participants were given detailed information about the study, including the purpose of the study and the responsibilities of the participants. Prior to data collection, ethical issues were discussed with participants, emphasizing confidentiality and protection from harm. Participants were encouraged to ask questions to address any concerns about their participation in the study. Informed consent was obtained from participants aged 18 years and above, while those below 18 years completed assent, in addition to parental informed consent.

The questionnaires were administered through face-to-face interviews conducted by trained research assistants in private rooms at the selected healthcare facilities. The research assistants were graduates from accredited tertiary institutions in Ghana. They completed a foundational course in psychology and behavioral sciences and were proficient in both English and Dangbe. They underwent comprehensive two-day training, covering interviewing techniques, ethical considerations in conducting research with vulnerable populations, and crisis intervention protocols. Interviews were conducted in each participant's preferred language (English or Dangbe), with each session lasting approximately 15–20 minutes.

Several quality assurance measures were implemented throughout the data collection process. The research team conducted daily reviews of completed questionnaires to ensure completeness and accuracy. Regular supervision and feedback sessions were held with research assistants, complemented by weekly team meetings to address challenges and maintain consistency in data collection procedures. For participants who exhibited signs of severe mental health concerns or suicidal ideation, immediate referral protocols were activated, with mental health professionals on standby at the facilities to provide necessary support and intervention. Following the completion of data collection, participants received reimbursement for their transportation costs to the healthcare institutions, with the total amount determined by the distance traveled. This approach ensured that participation in the study did not create a financial burden for the participants. Ethical approval was obtained from the Ghana Health Service Ethics Review Committee (GHS-ERC:004/07/24) and additional institutional permission was secured from the Eastern Regional Health Directorate and participating healthcare facilities

## Data analysis

Data were analyzed using IBM SPSS Statistics for Windows (version 27) (SPSS Inc, Chicago, IL, USA) and IBM SPSS AMOS 27 (IBM Corp., Armonk, NY, USA). Data were missing completely at random for less than 1% of cases with missing data on the study measures (Little's chi-square > 0.05). Consequently, the expectation-maximisation algorithm was used

to impute the missing data points. A series of confirmatory factor analysis was conducted to investigate the factor structure of the study measures using the maximum likelihood estimation method in Analysis of Moment Structures (AMOS) version 27. The fit of the models was evaluated using Chi-Square test and the following fit indicators [36]: comparative fit index (CFI; ≥ 0.90 = adequate; ≥ 0.95 = good), Tucker-Lewis index (TLI; ≥ 0.90 = adequate, ≥ 0.95 = good), the goodness of fit index (GFI; ≥ 0.90 adequate), root mean square error of approximation with its 90% confidence interval (RMSEA; 0.10 ≤ = acceptable. ≤ 0.08 = adequate, and ≤0.05 = good).

### Patient and public Involvement

The data underpinning the research was collected as part of an ongoing study designed to improve the mental health and psychosocial wellbeing of ALHIV and their caregivers. The entire project is underpinned by co-creation methodology involving the engagement of key stakeholders, including ALHIV and their caregivers, and healthcare professionals working in HIV clinics in the selected communities. These individuals have been instrumental in the study design and implementation, including highlighting the need to explore the utility and psychometric properties of the data collection measures. Their recommendation culminated into the examination of the psychometric properties of the study measures.

## Results

### Demographic characteristics of the participants

A total of 105 ALHIV participated in the study. More than half of the participants were females (73.3%) and attending school (78.1%). The majority were aged 10–19 years (59%), whereas 41% were aged 20–24 years. The duration of HIV diagnosis was 3 years (57.7%) and ≥ 4 years (42.3%).

### Factor Structure and Internal Consistency of Patient Health Questionnaire-9 (PHQ-9)

The goodness of fit indices for the various models of the PHQ-9 are summarized in Table 1.

   **One factor model.** In the first of the series of analyses, it was observed that the one factor model did not provide a good fit to the data ($\chi^2$ (27) = 87.32, p < .001; GFI = 0.84; TLI = 0.74; CFI = 0.81; RMSEA = 0.15). When the model was respecified by correlating the error variances of items 1 and 2 as well as 6 and 9, not all the fit indicators reached the recommended minimum threshold for a model fit ($\chi^2$ (25) = 37.59, p = .051; GFI = 0.95; TLI = 0.84; CFI = 0.89; RMSEA = 0.07). Nonetheless, the PHQ-9 items loaded satisfactorily on the "one-factor" PHQ-9, with the standard regression coefficients ranging from.45 (item 8) to.70 (item 2; See Table 2), and an internal consistency (Cronbach's

**Table 1. Summary of Chi Square Results and Goodness of Fit Indices for the various Measures.**

| Model | $\chi^2$ | df | $\chi^2$/df | GFI | TLI | CFI | RMSEA |
|---|---|---|---|---|---|---|---|
| **PHQ-9** | | | | | | | |
| One factor | 37.59 | 25 | 1.50 | .95 | .84 | .89 | .07 |
| Two factors | 47.10 | 25 | 1.88 | .92 | .90 | .93 | .09 |
| **GAD-7** | | | | | | | |
| One factor | 22.97 | 14 | 1.64 | .95 | .97 | .98 | .08 |
| Two factors | 25.13 | 13 | 1.93 | 0.94 | .96 | .98 | .09 |
| **Berger-10-item** | | | | | | | |
| One factor | 350.38 | 35 | 10.01 | .65 | .34 | .49 | .29 |
| Four factors | 117.96 | 30 | 3.93 | .84 | .79 | .86 | .17 |
| Three factors | 34.09 | 17 | 2.01 | .93 | .95 | .97 | .09 |

 

**Table 2. Study Measures and Their Corresponding Items and Factor Loadings.**

| S/N | Measures | Factor Loadings |
|---|---|---|
| | **Patient Health Questionnaire −9 (PHQ-9)** | |
| | **Cognitive/Affective subscale** | |
| 1 | Little interest or pleasure in doing things | 0.83 |
| 2 | Feeling down, depressed, or hopeless | 0.94 |
| 3 | Feeling bad about yourself — or that you are a failure or have let yourself or your family down | 0.55 |
| 4 | Thoughts that you would be better off dead or of hurting yourself in some way | 0.42 |
| | **Somatic subscale** | |
| 5 | Trouble falling or staying asleep, or sleeping too much | 0.67 |
| 6 | Feeling tired or having little energy | 0.54 |
| 7 | Poor appetite or overeating | 0.63 |
| 8 | Trouble concentrating on things, such as reading the newspaper or watching television | 0.49 |
| 9 | Moving or speaking so slowly that other people could have noticed? Or the opposite — being so fidgety or restless that you have been moving around a lot more than usual | 0.63 |
| | **Generalized Anxiety Disorder – 7 (GAD-7)** | |
| 1 | Feeling nervous, anxious, or on edge | 0.70 |
| 2 | Not being able to stop or control worrying | 0.94 |
| 3 | Worrying too much about different things | 0.90 |
| 4 | Trouble relaxing | 0.85 |
| 5 | Being so restless that it's hard to sit still | 0.70 |
| 6 | Becoming easily annoyed or Irritable | 0.26 |
| 7 | Feeling afraid as if something awful might happen | 0.59 |
| | **Berger's HIV Stigma Scale -Short Form (HSS-SF)** | |
| | **Personalized Stigma** | |
| 1 | I have been hurt by how people reacted to learning I have HIV. | 0.62 |
| 2 | I have stopped socializing with some people because of their reactions of my having HIV | 0.93 |
| 3 | I have lost friends by telling them I have HIV | 0.60 |
| | **Negative Self-Image** | |
| 4 | I feel that I am not as good a person as others because I have HIV. | 0.85 |
| 5 | Having HIV makes me feel unclean. | 0.94 |
| 6 | Having HIV makes me feel that I'm a bad person | 0.94 |
| | **Public Attitudes** | |
| 7 | Most people think that a person with HIV is disgusting. | 0.90 |
| 8 | Most people with HIV are rejected when others find out. | 0.92 |

Alpha) of 0.83. This means that although the regression coefficients and internal consistency of the PHQ-9 as one factor were adequate, the evidence did not support the PHQ-9 as a unidimensional measure of depressive symptoms.

**Two factor model.** The original two-factor model did not provide a good model fit, ($\chi^2$ (26) = 65.40, p < .001; GFI = 0.89; TLI = 0.83; CFI = 0.88; RMSEA = .12). Correlating the error variances of item 6 and 9 caused an improvement to the model: ($\chi^2$ (25) = 47.10, p = 0.005; GFI = 0.92; TLI = 0.90; CFI = 0.93; RMSEA = 0.09). The two factors (i.e., cognitive/affective symptoms and somatic factors) correlated significantly ($r$ = 0.71, p < 0.001). The items loaded satisfactorily onto their respect factors (See Table 2). The internal consistency of the cognitive/affective symptoms factor (4-item) was 0.79 whereas 0.70. The findings suggest that the PHQ-9 has a two-factor structure in the Ghanaian adolescents, comprising cognitive/affective symptoms and somatic symptoms, with adequate internal consistency. That is, the PHQ-9 measures two types of depressive symptoms- cognitive/affective symptoms and somatic, among ALHIV in Ghana.

### Factor structure and internal consistency of generalized anxiety disorder-7 (GAD-7)

The goodness of fit indices for the various models of the GAD-7 are summarized in Table 1.

**One factor model.** The original one factor model of the GAD-7 did not provide a good fit to the data ($\chi^2$ (14) = 66.54, p<.001; GFI=0.85; TLI=0.74; CFI=0.81; RMSEA=.19). However, after correlating the error variances of items 4 and 5, the resulting model showed a good fit ($\chi^2$(14) = 22.97, p=.042; GFI=0.95; TLI=0.97; CFI=0.98; RMSEA=.08). The items loaded satisfactorily on the GAD-7, with the standard regression coefficients ranging from 0.26 (item 6) to 0.94 (item 2) (Table 2). The internal consistency recorded was 0.88. The results showed that the GAD-7 can be represented as a unidimensional measure provided that the error variances of some of the items are allowed to correlate with each other.

**Two factor model.** The two-factor model exhibited excellent goodness of fit indices without any modification ($\chi^2$ (13) = 25.13, p=.022; GFI=0.94; TLI=0.96; CFI=0.98; RMSEA=0.09). The items loaded satisfactorily on their respective factors (See Table 2). The two factors correlated significantly (r=.85, p<0.001). The internal consistency for the cognitive/affective symptoms factor (4-item) was 0.86 and 0.66 for the somatic (3-item) factor. In summary, the GAD-7 measured two symptoms of anxiety among Ghanaian ALHIV, namely cognitive/affective symptoms and somatic symptoms. As a two-factor structure, the GAD-7 has good internal consistency, with the items loading satisfactorily onto their respective factors.

### Factor structure and internal consistency of berger hiv stigma scale-short form (HSS-SF)

**One-factor model.** The one factor model did not provide a good fit to the model, ($\chi^2$ (35) = 350.38, p<.001; GFI=0.65; TLI=.34; CFI=.49; RMSEA=.29).

**Four-factor model.** The model did not provide a good fit to the data ($\chi^2$ (30) = 117.96, p<.001; GFI=0.84; TLI=79; CFI=.86; RMSEA=.17).

**Three-factor model.** The four-factor model was respecified by removing the Disclosure factor due to poor correlation with other factors. The model that emerged following the re-specification provided a good fit to the data ($\chi^2$ (17) = 34.09, p=.008; GFI=0.93; TLI=0.95; CFI=0.97; RMSEA=0.09), as summarized in Table 1. The item loadings were satisfactory for the various factors, ranging from 0.60 (item 3) to 0.94 (item 7; Table 2). Internal consistency scores of 0.76, 0.94 and 0.95 were recorded for Personalized stigma, Negative self-image and Public attitudes subscales, respectively. It follows that the revised HSS-SF assesses three distinct type of stigma among ALHIV in Ghana, suggesting the HSS-SF could be better described as a three-factor model.

### Correlations among the study variables

The correlations between the study variables are summarized in Table 3. The PHQ-9 and GAD-7 subscales and total scales significantly correlated with each other, ranging from.50 to.96 (all *p*<0.01). In contrast, the Berger HSS-SF showed varying correlations. For example, only the total scale and Negative-Self-Image subscales correlated positively and significantly with the PHQ-9 and GAD-7 subscales and total scales, whereas the Public Attitude subscale did not.

### Discussions

The high mental health and stigma burden experienced by ALHIV necessitates efforts to screen and identify those at-risk for early intervention programming. The PHQ-9, GAD-7 and Berger HSS-SF are widely used to screen for depressive and anxiety symptoms, and stigma experiences, respectively. While these measures have the tendency to contribute to improving the mental health and well-being of ALHIV, it is crucial to interrogate their psychometric properties, especially the factor structure and internal consistency, which has remained the subject of contention [6].

The PHQ-9 has demonstrated a mixed factor structure in previous studies, with evidence for both one and two-factor structures [6]. The current study found evidence that the PHQ-9 in Ghanaian ALHIV is best represented by two-factors,

**Table 3. Summary of Correlation Between the Study Variables.**

| | 1 | 2 | 3 | 4 | 5 | 6 | 7 | 8 | 9 | 10 |
|---|---|---|---|---|---|---|---|---|---|---|
| **Depression-PHQ-9** | | | | | | | | | | |
| **1.**Cognitive/Affective | 1 | | | | | | | | | |
| **2.** Somatic | .61** | 1 | | | | | | | | |
| **3.** Total | .89** | .91** | 1 | | | | | | | |
| **Anxiety-GAD-7** | | | | | | | | | | |
| **4.** Cognitive/Affective | .65** | .46** | .62** | 1 | | | | | | |
| **5.** Somatic | .60** | .50** | .61** | .74** | 1 | | | | | |
| 6. Total | .67** | .51** | .66** | .96** | .91** | 1 | | | | |
| **Stigma-Berger's scale** | | | | | | | | | | |
| **7.** Personalized | .14 | −.03 | .06 | .26** | .12 | .22* | 1 | | | |
| **8.** Negative self-image | .45** | .19* | .35** | .41** | .27** | .38** | .19 | 1 | | |
| **9.** Public attitudes | .18 | .12 | .17 | .14 | .05 | .11 | .02 | .30** | 1 | |
| **10.** Total | .42** | .15 | .31** | .44** | .25* | .38** | .62** | .82** | .53** | 1 |
| **Mean** | 2.88 | 2.87 | 5.74 | 3.83 | 2.65 | 6.48 | 6.07 | 7.04 | 7.78 | 20.88 |
| **SD** | 1.5 | 1.4 | 1.33 | 1.40 | 1.41 | 2.43 | 2.89 | 4.35 | 2.28 | 6.88 |
| **Skewness** | 1.23 | 1.48 | 1.50 | .65 | .62 | .62 | .77 | .71 | −.47 | .46 |
| **Kurtosis** | .44 | .42 | .20 | −.41 | −.48 | −.33 | −.58 | −.96 | −.95 | −.45 |
| **Minimum value** | 0 | 0 | 0 | 0 | 0 | 0 | 3 | 3 | 2 | 8 |
| **Maximum value** | 12 | 15 | 27 | 0 | 9 | 21 | 15 | 15 | 10 | 39 |

**Correlation is significant at the 0.01 level (2-tailed).

*Correlation is significant at the 0.05 level (2-tailed).

consisting of cognitive/affective and somatic domains. These two factors correlated significantly and demonstrated good internal consistency. These observations are largely consistent with the findings of previous studies [6], while contrasting others [8,16]. All in all, our study proposes that for ALHIV, it would be beneficial to explore their affective/cognitive and somatic domains separately to obtain robust information about their mental health and wellbeing to inform both research and clinical practice. Stated alternatively, the PHQ-9 should be seen as assessing different categories of depressive symptoms, namely cognitive/affective symptoms and somatic symptoms, given us a better representation and distribution of the depressive tendencies among ALHIV. For a nuanced understanding of depressive symptoms, the study did not support previous endorsement of a unidimensional factor structure of the PHQ-9, although the scores for both cognitive/affective symptoms and somatic domains can be summed to obtain total score on depression. The foregoing discussion is important given that affective, cognitive, and somatic symptoms are critical components for diagnosis of depression, hence the need to treat the PHQ-9 as a two-factor measure [8]. The GAD-7, just like the PHQ-9, was found to comprise two factors, namely cognitive/affective and somatic symptoms, with good internal consistency. It can also be represented as a unidimensional factor, supporting the bulk of existing studies [7,23,24]. The PHQ-9 and GAD-7 correlated significantly with each other, suggesting that the presence of anxiety symptoms signals depressive symptoms, and vice versa. The PHQ-9 and GAD-7 items loaded satisfactorily onto their respective factors, with all factor loadings exceeding the conventional value of 0.30 [37]. The difference in findings between this and previous studies could be attributed to methodological issues, including sample size and characteristics, mode of data collection (i.e., self-report versus interviewer administered), and more crucially the dynamic influence of sociocultural factors on the assessment of psychopathologies.

Furthermore, the two-factor structure of PHQ-9 identified in our Ghanaian sample may reflect culture-specific expressions of depression among ALHIV in West African contexts. In collectivistic cultures like Ghana, mental health symptoms often manifest through somatic complaints as well as direct expression of emotional distress, which may explain the

distinct cognitive/affective and somatic domains found in our analyses [8]. This finding aligns with the research indicating that cultural factors could impact how mental health symptoms are experienced and reported across different populations [38]. The sociocultural context of Ghana, characterized by strong family ties, religious beliefs, and communal living arrangements, creates a unique environment in which stigma and mental health challenges manifest. Ghanaian ALHIV often experience stigma shaped by cultural beliefs about illness causation, moral judgments, and fears of contagion that may differ from those in Western settings. The measurement of these experiences requires culturally sensitive approaches that acknowledge these distinctions. The PHQ-9 and GAD-7 have proven useful in this regard as they demonstrated good factorial structure, internal consistency and correlations among ALHIV in Ghana.

The Berger HSS-SF was derived from the Berger HSS using exploratory factor analysis, as stated previously. To the best of our knowledge, this is the first attempt to establish the factor structure of the Berger HSS-SF using CFA. Although one-factor and the original four-factor structure were not established, a three-factor structure was successfully confirmed following the deletion of the Disclosure subscale from the analysis. The three factors were Personalized stigma, Negative self-image and Public attitudes. The item loadings for each factor exceeded the conventional value of 0.30 [37]. The factors also demonstrated good internal consistency; however, they showed poor or no significant correlation with the PHQ-9 and GAD-7 subscales and total scales. The poor utility of the Disclosure factor in this study points to the complexity and difficulty surrounding HIV disclosure discourses in Ghanaian cultures, where discussing one's HIV status may have more severe social consequences than in more individualistic societies. In Ghana, family support systems are central to adolescents' coping mechanisms, but familial stigma can also be profound when HIV status is revealed. This cultural paradox may explain why the disclosure component of stigma operates differently in this population compared to Western contexts where the Berger scale was originally developed. Furthermore, spiritual beliefs about illness causation in Ghanaian culture add another layer to stigma experiences that may not be fully captured by Western-developed instruments. A weak to no correlation was observed between the factors of the Berger HSS-SF. These together raise initial questions about the utility of the Berger HSS-SF as a short form of the Berger HSS in the Ghanaian cultural context. Additional studies are needed to investigate the psychometric properties of the Berger HSS-SF within various African cultural settings, with particular attention to how cultural-specific expressions of stigma may necessitate adaptation of measurement tools.

## Clinical implications of the study

In Ghana and several other African countries, there are growing calls and concerted efforts to integrate mental health into primary healthcare for all categories of patients, including ALHIV. The availability of validated and psychometrically sound measures will contribute significantly towards the realization of integrated healthcare, by facilitating screening for mental health issues for early intervention. The study contributed to this growing agenda by providing preliminary evidence supporting the factor structure and reliability of the PHQ-9, GAD-7 and Berger HSS-SF among ALHIV. To proceed in this regard, healthcare professionals can be trained to administer the PHQ-9, GAD-7 and Berger HSS-SF as part of the routine healthcare services to ALHIV to support evidence-based decision making such as recommendations for detailed assessment and intervention programming to improve their mental health and wellbeing. Early screening and detection would support early intervention programming to address subliminal mental health problems from developing into full-blown, diagnosable mental health disorders among ALHIV.

## Limitations of the study

The study findings should be examined considering the following limitations. The sample size was small, although it satisfies the requirement for CFA by Kline [39] who recommended 10:1 respondent-to-item ratio for psychometric investigation. More crucially, the findings observed for the PHQ-9 and GAD-7 were consistent with those reported by previous studies with relatively large sample size. The Berger HSS-SF may demonstrate different factorial structure, perhaps with a large sample size. The uneven distribution of the participants based on gender (28 males versus 77 females) also restricted

gender invariance testing, thereby limiting discussions on the utility of the measures for gender-specific screening. The lack of comparison groups (e.g., HIV-negative adolescents) restricts our ability to determine whether the observed factor structures are specific to ALHIV or reflect broader adolescent mental health patterns in Ghana.

Third, the absence of test-retest reliability data limits our understanding of measurement stability over time and the lack of concurrent validity measures against clinical diagnoses restricts our understanding of these instruments' diagnostic utility. The data collection methods provided no mechanisms to authenticate the responses of the participants, creating the possibility for underreporting or over-reporting of symptoms of depression, anxiety and stigma. The reliance on self-report measures and face-to-face interviews may have introduced social desirability bias, particularly given the sensitive nature of questions about mental health and stigma experiences. Indeed, the Ghanaian society and cultures stigmatize HIV, culminating into discriminatory practices and other negative behaviors. These experiences could potentially bias their responses. Notwithstanding the limitations above, the preliminary findings reported on the factor structure of the PHQ-9 and GAD-9 among ALHIV in Ghana are consistent with previous studies (Stanyte et al., 2023), thereby giving some credence to the study.

## Conclusion

The study has contributed to the cross-cultural literature regarding the administration of the PHQ-9, GAD-7 and Berger HSS-SF among ALHIV in Ghana. The study lends support to the discussion on the cross-country application of behavioral measures. We conclude that PHQ-9 is better represented as a two-factor structure, consisting of cognitive/affective symptoms and somatic behaviors, whereas the GAD-7 is a unidimensional measure of anxiety symptoms. The depressive and anxiety symptoms measured by the PHQ-9 and GAD-7 can be generalized to different cultures and population. The Berger HSS-SF was adequately represented by a three-factor structure instead of the original four factors. The deletion of the Public Disclosure subscale improved the factor structure of the Berger HSS-SF. The finding raises questions about the utility of the Public Disclosure in ALHIV discourse, necessitating the call for future research.

## Supporting information

**S1 File. Questionnaires.**
(DOCX)

## Acknowledgments

The authors thank the adolescents and their families who contributed to this study. We also thank the management and staff of Atua Government Hospital, Akuse Government Hospital, and Asesewa Government Hospital for their invaluable support during data collection. Special appreciation goes to the research assistants who helped with data collection

## Author contributions

**Conceptualization:** Samuel Adjorlolo.

**Data curation:** Dorothy Serwaa Boakye.

**Formal analysis:** Samuel Adjorlolo.

**Funding acquisition:** Samuel Adjorlolo.

**Investigation:** Dorothy Serwaa Boakye.

**Methodology:** Dorothy Serwaa Boakye.

**Project administration:** Samuel Adjorlolo.

**Supervision:** Samuel Adjorlolo.

**Writing – original draft:** Samuel Adjorlolo.

**Writing – review & editing:** Samuel Adjorlolo, Dorothy Serwaa Boakye.

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
