## [Decision Letter · Decision Letter 0]

5 May 2025-05-05

Dear Dr. Adjorlolo,

Thank you for submitting your manuscript to PLOS ONE. After careful consideration, we feel that it has merit but does not fully meet PLOS ONE’s publication criteria as it currently stands. Therefore, we invite you to submit a revised version of the manuscript that addresses the points raised during the review process.

We look forward to receiving your revised manuscript.

Kind regards,

Paweł Larionow, Ph.D.

Academic Editor

PLOS ONE

“The study was made possible (in-part) by the Preparing Outstanding Social Science Investigators to Benefit Lives and Environments in Africa Initiative (POSSIBLE-Africa) (Grant NO# POS-24-07), an initiative of the Science for Africa Foundation (SFA Foundation) enabled by the support of Carnegie Corporation of New York. The statements made and the views expressed are solely the responsibility of the authors and necessarily those of the SFA Foundation and partners.”

“The study was made possible (in-part) by the Preparing Outstanding Social Science Investigators to Benefit Lives and Environments in Africa Initiative (POSSIBLE-Africa) (Grant NO# POS-24-07), an initiative of the Science for Africa Foundation (SFA Foundation) enabled by the support of Carnegie Corporation of New York. The statements made and the views expressed are solely the responsibility of the authors and necessarily those of the SFA Foundation and partners.”

Please include your amended Funding Statement within your cover letter. We will change the online submission form on your behalf."

Reviewers' comments:

Reviewer's Responses to Questions

**Comments to the Author**

1. Is the manuscript technically sound, and do the data support the conclusions?

Reviewer #1: Yes

Reviewer #2: Yes

2. Has the statistical analysis been performed appropriately and rigorously?

Reviewer #1: Yes

Reviewer #2: Yes

3. Have the authors made all data underlying the findings in their manuscript fully available?

Reviewer #1: Yes

Reviewer #2: Yes

4. Is the manuscript presented in an intelligible fashion and written in standard English?

Reviewer #1: Yes

Reviewer #2: Yes

Reviewer #1: 1. It is recommended that the discussion section further address cultural differences in the experience of stigma and mental health.

2. It is also suggested that measurement invariance across subgroups (e.g., age or gender) be examined to enhance the generalizability of the findings.

3. Concurrent validity with clinical measures is also recommended for future studies.

4. Overall, this manuscript represents an excellent example of instrument validation in a distinct cultural context and could serve as a valuable regional reference.

Reviewer #2: REVIEW REPORT: Factorial structure of the Patient Health Questionnaire -9(PHQ-9) Generalized Anxiety Disorder-7(GAD-7) and Berger HIV Stigma Scale-Short Form (HSS-SF) among Adolescents living with HIV in Ghana

The paper is a quantitative study entitled ‘Factorial structure of the Patient Health Questionnaire -9(PHQ-9) Generalized Anxiety Disorder-7(GAD-7) and Berger HIV Stigma Scale-Short Form (HSS-SF) among Adolescents living with HIV in Ghana ‘This title is a bit too long. It would be better if the authors avoided abbreviations in the title but have them in the main text.

Abstract

The abstract is well detailed and addresses the main research question. The study addressed one main objective namely; the factor structure, internal consistency, and correlations of the PHQ-9, GAD-7, and HSS-SF among ALHIV in Ghana. This is well captured in the abstract.

Introduction

The introduction and background information are nicely detailed and has clearly identified the gap in the usage of the 3 tools.

Materials and Methods

The study design, setting, exclusion and inclusion criteria are well explained but the sampling procedure is unclear. In line 174-175, the authors should state, out of the total percentage of the required population what percentage was 63 for Atua, 16 for Akuse and 26 for Asesewa. Being a quantitative study it is important to explain how the sample size was arrived at. The scope of the study is well provided and the presentation is consistent throughout the manuscript.

Data collection procedure

This section is detailed and well written.

Data analysis and Results

• The findings that the study produces are very critical but have no implications, and they only need one or two sentences to bring out the implication of each finding so that the study remains in context.

• The manuscript is well written with key subtopics availed. The authors however need to be more analytical in data presentation. The results are presented and not interpreted at all. Making it difficult to make exact meaning. It is always good practice to interpret all findings, this makes it easy to come up with the implications of the findings. The manuscript would be more useful to a broader readership if the authors moved from just providing results to interpreting them and giving the implication as well.

• I tend to think that the authors oversimplified the findings thus inhibiting the deeper message that the subject under study could provide.

• All results presented in tables need some write up to explain the figures rather than simply putting the table and leaving them that way. All tables must have some explanation.

Discussion

On the PHQ-9 the authors propose that it would be beneficial to explore the affective/ cognitive and somatic domains separately (Line 347). How about separating cognitive and affective domains or why do the authors think that cognitive and affective should be studied together in the case of Ghana. Generally, the discussion is good but as a wrap up what was the most common finding of this study and what research gap does it fill? Or what new information came from this study? The discussion section should be improved by linking the study findings to earlier works on the subject area and explaining how the current study complements previous ones.

Limitations

What about the sociocultural influences on the responses of the adolescents? Don’t you think this is a limitation?

Conclusion

The conclusion is good except line 401-402-what did HSS-SF demonstrate? It would be nice to state the findings before indicating that they are not sufficient and acceptable.

References

Well done, only needs uniformity. Choose one style of referencing and use it – do not mix.

**Do you want your identity to be public for this peer review?** For information about this choice, including consent withdrawal, please see our Privacy Policy

Reviewer #1: No

Reviewer #2: No

---

## [Editor Report · Decision Letter 1]

12 May 2025

Dear Dr. Adjorlolo,

Thank you for submitting your manuscript to PLOS ONE. After careful consideration, we feel that it has merit but does not fully meet PLOS ONE’s publication criteria as it currently stands. Therefore, we invite you to submit a revised version of the manuscript that addresses the points raised during the review process.

*M* , *SD*

We look forward to receiving your revised manuscript.

Kind regards,

Paweł Larionow, Ph.D.

Academic Editor

PLOS ONE
---

## [Editor Report · Decision Letter 2]

27 May 2025

Factorial Structure of the Patient Health Questionnaire-9, Generalized Anxiety Disorder-7 and Berger HIV Stigma Scale-Short Form among Adolescents Living with HIV in Ghana.

PONE-D-25-17680R2

Dear Dr. Adjorlolo,

We’re pleased to inform you that your manuscript has been judged scientifically suitable for publication and will be formally accepted for publication once it meets all outstanding technical requirements.

Kind regards,

Paweł Larionow, Ph.D.

Academic Editor

PLOS ONE
---

## [Editor Report · Acceptance letter]

PONE-D-25-17680R2

PLOS ONE

Dear Dr. Adjorlolo,

I'm pleased to inform you that your manuscript has been deemed suitable for publication in PLOS ONE. Congratulations! Your manuscript is now being handed over to our production team.

Kind regards,

on behalf of

Dr. Paweł Larionow

Academic Editor

PLOS ONE